# Prospective individual patient data meta-analysis of two randomized trials on convalescent plasma for COVID-19 outpatients

Pere Millat-Martinez[1,2,62], Arvind Gharbharan[3,62], Andrea Alemany[1,4,5,62], Casper Rokx [3], Corine Geurtsvankessel [6], Grigorios Papageorgiou[7], Nan van Geloven [8], Carlijn Jordans[3], Geert Groeneveld[9], Francis Swaneveld[10], Ellen van der Schoot[11], Marc Corbacho-Monné[1,5,12], Dan Ouchi [1,13], Francini Piccolo Ferreira [14], Pierre Malchair[15], Sebastian Videla [1,16,17], Vanesa García García[15], Anna Ruiz-Comellas [18,19,20], Anna Ramírez-Morros [18], Joana Rodriguez Codina[21], Rosa Amado Simon[21], Joan-Ramon Grifols[4,22], Julian Blanco [23], Ignacio Blanco [4,24], Jordi Ara[4,25], Quique Bassat [2,26,27,28,29], Bonaventura Clotet[1,4,20,23], Bàrbara Baro [2], Andrea Troxel[30], Jaap Jan Zwaginga[31,32,63], Oriol Mitjà[1,4,20,33,63], Bart J. A. Rijnders [3,63 ✉], CoV-Early study group* & COnV-ert study group*

Data on convalescent plasma (CP) treatment in COVID-19 outpatients are scarce. We aimed to assess whether CP administered during the first week of symptoms reduced the disease progression or risk of hospitalization of outpatients. Two multicenter, double-blind randomized trials (NCT04621123, NCT04589949) were merged with data pooling starting when <20% of recruitment target was achieved. A Bayesian-adaptive individual patient data meta-analysis was implemented. Outpatients aged ≥50 years and symptomatic for ≤7days were included. The intervention consisted of 200–300mL of CP with a predefined minimum level of antibodies. Primary endpoints were a 5-point disease severity scale and a composite of hospitalization or death by 28 days. Amongst the 797 patients included, 390 received CP and 392 placebo; they had a median age of 58 years, 1 comorbidity, 5 days symptoms and 93% had negative IgG antibody-test. Seventy-four patients were hospitalized, 6 required mechanical ventilation and 3 died. The odds ratio (OR) of CP for improved disease severity scale was 0.936 (credible interval (CI) 0.667–1.311); OR for hospitalization or death was 0.919 (CI 0.592–1.416). CP effect on hospital admission or death was largest in patients with ≤5 days of symptoms (OR 0.658, 95%CI 0.394–1.085). CP did not decrease the time to full symptom resolution.
Trial registration: Clinicaltrials.gov NCT04621123 and NCT04589949. Registration: NCT04621123 and NCT04589949 on https://www.clinicaltrials.gov

A full list of author affiliations appears at the end of the paper.

The unprecedented pace and amount of research on the pathogenesis of severe acute respiratory syndrome coronavirus 2 (SARS-CoV-2) led to the availability of mortality-reducing therapies within a year after the start of the coronavirus disease 2019 (COVID-19) pandemic[1–3]. For non-hospitalized COVID-19 patients, only anti-SARS-CoV-2 monoclonal antibodies have emerged as a treatment that reduces hospital admission but only when given in the first week of illness. However, they are typically unavailable to middle and low-income countries[4–7].

Convalescent plasma (CP) from COVID-19 recovered patients contains polyclonal anti-SARS-CoV-2 antibodies, can be collected in large quantities at relatively low costs and was used as a therapeutic strategy in previous viral outbreaks[8,9]. So far, randomized trials were unable to generate convincing evidence in support of CP for hospitalized patients with COVID-19[10–18]. However, because an autologous SARS-CoV-2 antibody response typically precedes hospital admission, CP is more likely to be beneficial when it is administered very early after symptom onset[19]. Indeed, the only evidence from a randomized trial in favor of CP for COVID-19 comes from a small study in which elderly outpatients received CP in the first 72 h after symptom onset[20]. In a more recent trial, CP did not reduce the risk of disease progression of COVID-19 in patients with early disease (≤7 days). However, in this trial, patients were recruited at emergency rooms and were, therefore, more likely to manifest severe symptoms[21]. This approach resulted in a trial profile of patients with moderate or late-stage disease, opposed to what was intended in the design. Hence, whether early treatment with CP improves the outcome of outpatients with COVID-19 remains an important question.

As soon as effective vaccines against COVID-19 became available in high-income countries, they were prioritized for individuals at higher risk for a poorer COVID-19 outcome. Because studies on CP for outpatients with COVID-19 focus on these high-risk populations as well, a high vaccination uptake will reduce the number of COVID-19 patients eligible for these studies. More importantly, the risk for a severe outcome will be small when patients become infected despite vaccination. Therefore, we anticipated that vaccination would slow down recruitment, reduce the number of events in the recruited patients and result in individual studies being underpowered. In light of the uncertainty for achieving recruitment goals, real-time pooling of individual patient data from ongoing clinical trials was proposed as a tool for providing timely data to respond to the public health crisis[22]. With this in mind, we initiated the COntinuous Monitoring of Pooled International trials of convaLEscent plasma for COVID-19 patients at home Consortium (COMPILE_home), which provided a platform to pool individual patient data continuously and in real-time from randomized clinical trials (RCTs) on CP for outpatients with COVID-19[22]. This COMPILE_home consortium prospectively pooled and monitored the data from 2 double-blind RCTs, the CoV-Early (NCT04589949) and the COnV-ert (NCT04621123) studies, to assess the effectiveness of high-titer CP for COVID-19 outpatients.

## Results

**Trials profile.** The search for trials resulted in 35 identified studies, thirty-one of which did not meet the selection criteria of the consortium (Supplementary Fig. 1). Of the four remaining studies, one study team opted to abstain from pooling data while another never responded to repeated emails and calls, resulting in two trials included in the pooled analysis: The COnV-ert study (NCT04621123) and the CoV-Early study (NCT04589949). The COnV-ert study received approval from the Institutional Review Board of the Hospital Germans Trias I Pujol (reference PI 20-313) and the CoV-Early study received approval from the Institutional Review Board of the Erasmus Medical Center Rotterdam (reference MEC-2020-0682). Briefly, the COnV-ert study randomized outpatients at 4 sites in Catalunya (Spain) aged ≥50 years with ≤7 days of symptoms to one unit (200–300 mL) of CP or sterile 0.9% saline solution, both covered with opaque tubular bags for blinding investigators and patients. The COnV-ert study joined the consortium when 65 of 474 planned patients were enrolled. CoV-Early enrolled outpatients at 10 sites aged ≥50 years with ≤7 days of symptoms and at least one additional risk factor for severe COVID-19 to receive either one unit (300 mL) of CP or non-convalescent plasma (donated before 01/2020) masked to investigators and patients. It had randomized 150 of the 690 planned patients when they joined the consortium. Details about the allocation concealment, blinding and selection of CP donors in both trials can be found in Supplementary Table 2 and the study protocols.

The COnV-ert study selected the CP after being screened for high anti-SARS-CoV-2 IgG titers with ELISA (EUROIMMUN ratio ≥6), according to guidelines, and supplied by the regional blood bank (Banc de Sang i Teixits de Catalunya—BST); and the CoV-Early study selected the convalescent plasma based on a plaque reduction neutralization test (PRNT) 50 titer of 1:160 or higher. The two trials used a different assay to measure the titer of SARS-CoV-2 neutralizing antibodies. Therefore, a panel of 15 plasma samples was provided for comparison by the Support-E consortium, aimed at harmonizing CP evaluation in Europe[23]. These results confirmed the linearity of both assays and allowed conversion of all neutralizing antibody titers into international units (IU/mL). The median neutralizing antibody titer in the plasma units was 1:386 (IQR 1:233–1:707) IU/mL, which is twice the median titer we previously observed in Dutch CP donors[19]. More details are described in the online Supplementary Data (page 13), in Supplementary Tables 1, 2 and in the individual study protocols.

**Study patients and recruitment.** Between November 2020 and July 2021, the CoV-Early and COnV-ert study teams contacted approximately 4450 outpatients with a positive SARS-CoV-2 PCR or an antigen test. The majority of exclusions occurred for one of the following reasons: few remaining or clearly improving symptoms, no comorbidities, >7 days of symptoms, unable to come to study site or declined to participate. The online supplement provides more information about the recruitment procedures of each trial.

The rapid uptake of COVID-19 vaccination in Europe, which significantly affected recruitment rate in both studies (Supplementary Fig. 2) and the authorization of specific anti-SARS-CoV-2 monoclonal antibodies for high-risk outpatients resulted in early trial termination (COnV-ert on 8th of June and CoV-Early on 13th of July 2021) following recommendations of their DSMBs. By that time, 797 participants had been enrolled and 782 of them had received the allocated intervention and could be pooled for the analysis (Fig. 1).

Patients included in the analysis had a median age of 58 years (IQR 53–64), a median of 5 days (IQR 4–6) from symptom onset, and a median of 1 comorbidity (IQR 0-2). According to the baseline assessment, 688 patients (93%) had a negative result for serum IgG anti-SARS-CoV-2 S-protein, and 21 had completed their COVID-19 vaccination. 14 participants had received one of 2 doses of a mRNA vaccine at the time of inclusion. Baseline characteristics were comparable between both study arms (Table 1).

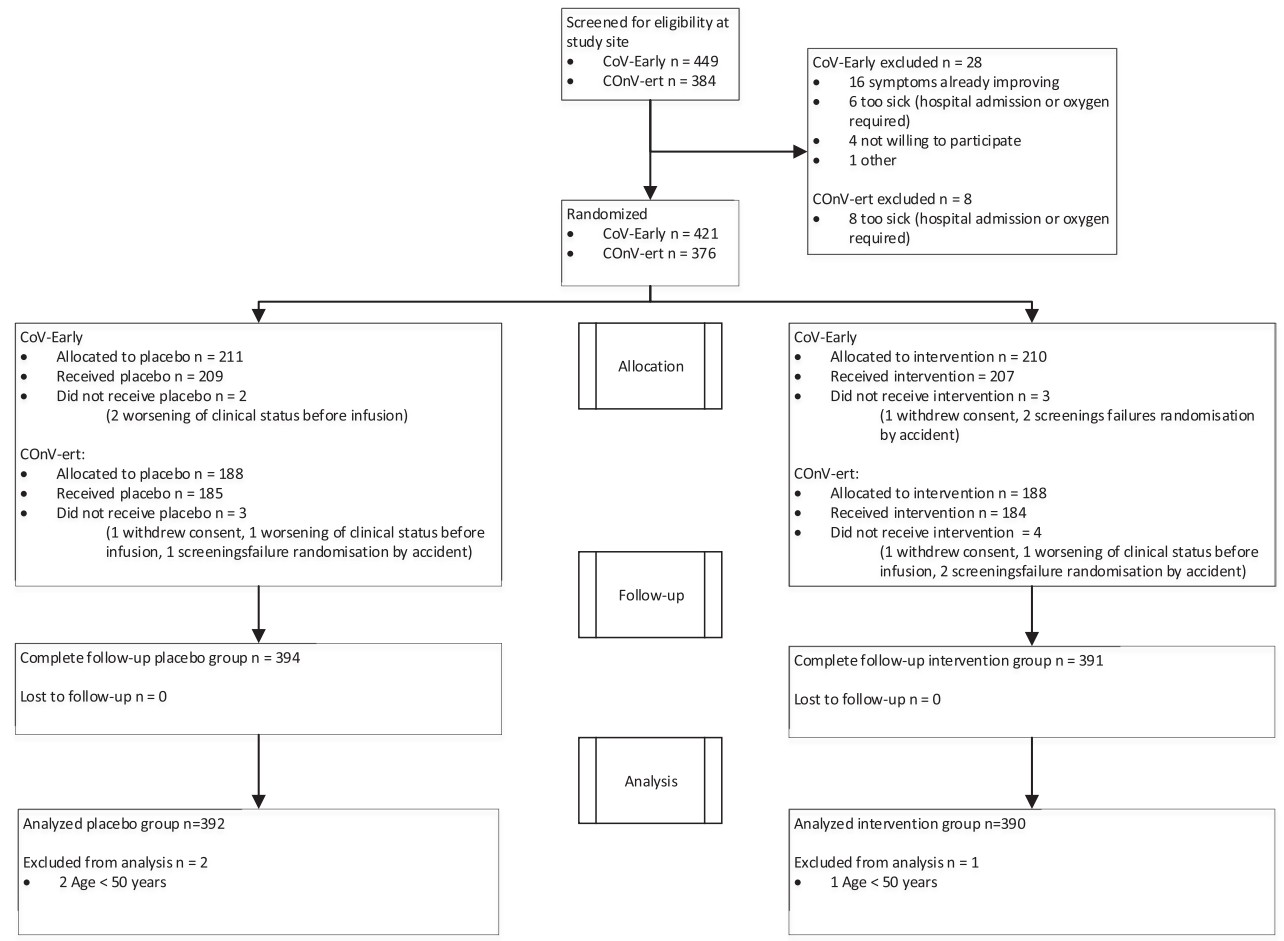

**Fig. 1 CONSORT flow diagram.** Figure shows the CONSORT flow diagram of the COMPILEhome patients. 833 patients were screened at a study site and 782 were included for analysis.

**Table 1 Baseline characteristics.**

| Characteristic | Total (*n* = 782) | CP[a] (*n* = 390) | Control (*n* = 392) |
|---|---|---|---|
| Male sex—no. (%) | 522 (66.8%) | 267 (68.5%) | 255 (65.1%) |
| Age—median (IQR) | 58 (53-64) | 58 (53-64) | 58 (54-65) |
| 50–60 y | 428 | 222 | 206 |
| 61–70 y | 217 | 103 | 114 |
| >70 y | 82 | 36 | 46 |
| O$_2$ saturation—median (IQR)[b] | 97 (96-98) | 97 (96-98) | 97 (96-98) |
| Severe immunodeficiency—no. (%) | 13 (1.7%) | 5 (1.3%) | 8 (2.1%) |
| Number of comorbidities—median (IQR)[c] | 1 (0-2) | 1 (0-2) | 1 (0-2) |
| 0 | 225 | 111 | 114 |
| 1 | 349 | 171 | 178 |
| 2–3 | 192 | 100 | 92 |
| >3 | 15 | 7 | 8 |
| Days since first symptoms—median (IQR) | 5 (4-6) | 5 (4-6) | 5 (4-6) |
| Positive antibody status at baseline—no. (%) | 53 (7.0%) | 28 (7.7%) | 24 (6.4%) |

[a]Convalescent plasma.
[b]Baseline oxygen saturation without supplementary oxygen.
[c]Obesity, cardiac disease, lung disease, neurological disease, diabetes, chronic renal failure, cancer and/or liver disease. See the Supplementary Appendix for additional details of the comorbidities.

**Primary endpoints**. Table 2 and Fig. 2 show the distribution of patients across the five categories of the disease severity scale. The overall estimated OR for patients treated with CP was 0.936 (posterior mean, 95% credible interval 0.667–1.311) with a 64.9% posterior probability of benefit (OR <1). Hospital admission or death occurred in 34 of 390 (8.7%) patients treated with CP and in 40 of 392 (10.2%) patients in the control arm with an OR of 0.919 (posterior mean, 95% credible interval 0.592–1.416) and a 64.3% posterior probability of benefit. Although being included in the COnV-ert trial was associated with a poorer overall outcome,

**Table 2 Distribution of the outcome of the patients in the 28 days after inclusion across the 5-points disease severity scale.**

| Worst disease severity score | Total (n = 782) | CP[a] (n = 390) | Control (n = 392) |
|---|---|---|---|
| Recovered before day 8 after transfusion—no. (%)[b] | 143 (18.3%) | 74 (19.0%) | 69 (17.6%) |
| Continued symptoms after day 7—no. (%)[c] | 565 (72.3%) | 282 (72.3%) | 283 (72.2%) |
| Admitted to hospital but no invasive ventilation needed—no. (%) | 65 (8.3%) | 31 (7.9%) | 34 (8.7%) |
| Admitted to hospital and invasive ventilation needed—no. (%) | 6 (0.8%) | 2 (0.5%) | 4 (1.0%) |
| Death—no. (%) | 3 (0.4%) | 1 (0.3%) | 2 (0.5%) |

[a]Convalescent plasma.
[b]Recovered with no symptoms related to COVID-19 within 7 days after inclusion.
[c]Continued symptoms attributable to COVID-19.

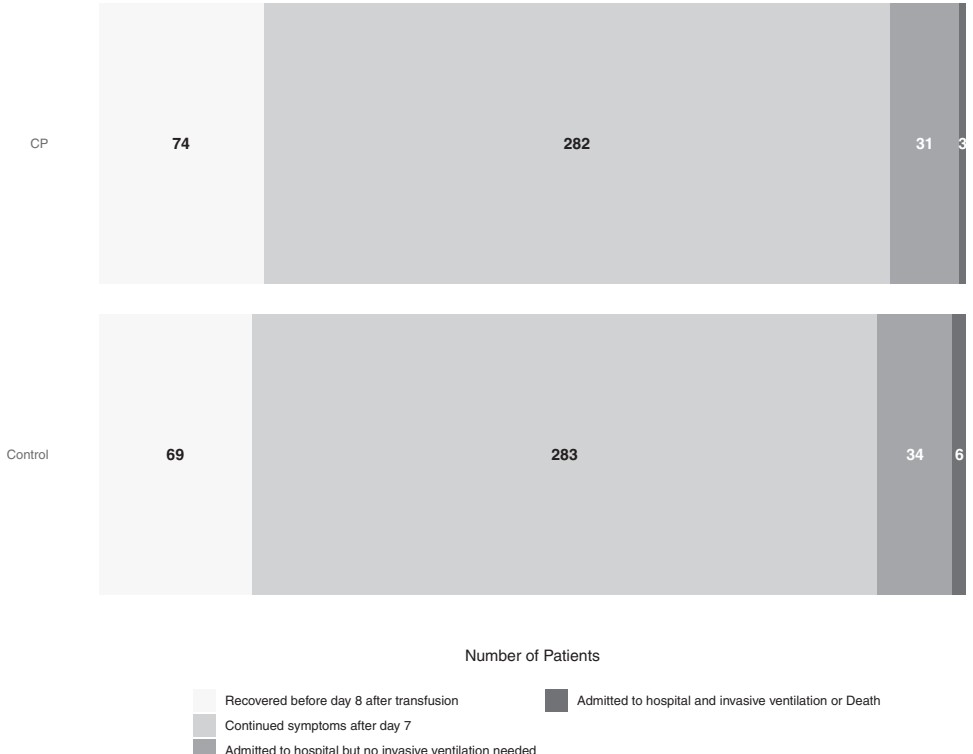

**Fig. 2 Distribution for COVID-19 severity at 28 days.** CP Convalescent plasma. Figure shows the distribution of the outcome of the patients in the 28 days after inclusion across the 5-point disease severity scale: 1 = recovered before day 8 after transfusion, 2 = continued symptoms after day 7, 3 = hospital admission, 4 = invasive ventilation, 5 = death. Moving from lighter to darker shading represents increasing scores on the severity scale. The darker shade includes point 4 and 5 of the scale (invasive ventilation or death).

the effect of CP was similar in both trials. This increased risk for patients in COnV-ert was independent of age, sex, and the number of comorbidities. The results of all covariates included in the primary analysis can be found in the online supplement (Supplementary Figs. 5, 6).

**Secondary endpoints.** No differences between CP and control patients regarding time to complete resolution of COVID-19 symptoms was seen (log-rank p = 0.66, Fig. 3). The effect size of CP on the binary outcome of hospital admission or death was larger in patients with ≤5 days of symptoms (OR 0.658, 95% CI 0.394-1.085) compared to those with >5 days (OR 1.427, 95% CI 0.789–2.580) and comparable results were observed for the ordinal outcome (OR 0.720, 95% CI 0.486–1.064 in the early treated group, Supplementary Figs. 7, 8).

Finally, the ORs for patients who received CP with neutralizing antibody titers above or below the median titer were nearly identical (Supplementary Fig. 9). Also, no notable difference was

observed when patients with IgG anti-SARS-CoV-2 antibodies detected at baseline were excluded (OR 0.880, 95% CI 0.590–1.310 for the binary outcome, OR 0.892, 95% CI 0.643–1.236 for ordinal outcome, Supplementary Figs. 7, 8).

**Safety.** The intervention was well-tolerated. 89 serious adverse events (SAE) were reported, 4 were considered related to the plasma transfusion (3 in the control arm). Three patients could leave the hospital <24 h after transfusion while the fourth was hospitalized for 5 days 1 week after the CP transfusion and diagnosed with thrombophlebitis at the infusion site and a pulmonary embolism (Table 3).

**Discussion**

In this analysis of 782 patients with COVID-19 randomized to high-titer CP or placebo within 7 days of disease onset, treatment with CP did not prevent COVID-19 progression, hospitalization, or other clinical outcomes. Our results agree with those by Korley

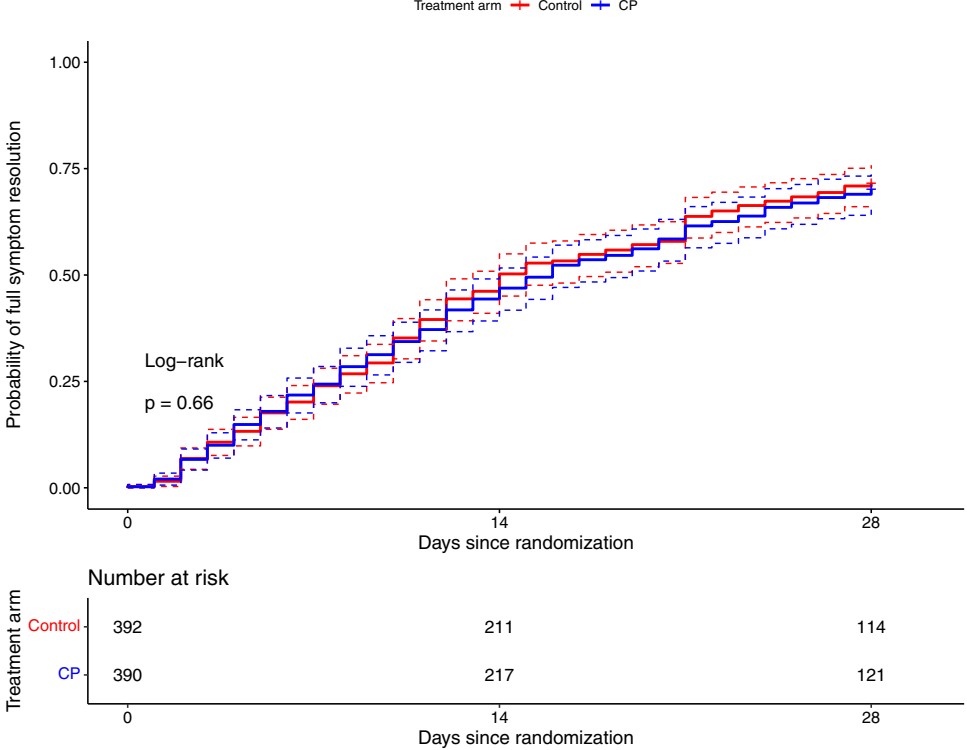

**Fig. 3 Time to full symptom resolution up to day 28 (end of follow-up).** CP Convalescent plasma. Log-rank test *p* = 0.66. The dotted error bars represent the 95% CI.

**Table 3 Serious adverse events[a].**

| SAE category | Total | CP[b] | Control |
|---|---|---|---|
| (Prolongation of) hospital admission—no.[c] | 80 | 37 | 43 |
| Death—no. | 3 | 1 | 2 |
| Serious transfusion related adverse event—no.[d] | 4 | 1 | 3 |
| Life threatening transfusion reaction—no.[e] | 2 | 0 | 2 |
| Other AE | 2 | 1 | 1 |

[a]Serious adverse events (SAE) were registered in all patients that signed the informed consent form (*n* = 797) regardless of being transfused or not.
[b]Convalescent plasma.
[c]When a patient is hospitalized more than once, each admission is counted separately.
[d]Any transfusion reaction associated with a plasma transfusion that was considered as a SAE.
[e]2 patients with anaphylaxis very soon after discharge that required urgent therapy by paramedics.

et al. in patients of the same age and symptom duration but with probably more severe symptoms as they were recruited at emergency rooms in the USA[21]. These findings differ from those of a smaller trial that used CP within 72 h of symptom onset in much older patients (≥75 years)[20]. We explored signs of efficacy in various subgroups most likely to benefit from CP. The only subgroup in our study that we found that could potentially benefit from CP was the subgroup with ≤5 days from the onset of symptoms (OR 0.70, CI 0.47–1.03). The potential effect of CP when administered early after disease onset has been suggested by other authors[24], and could explain the results reported by Libster et al. study[20]. However, in our study this was a secondary end-point and the confidence interval was wide, so confirmation in other studies is needed. Regarding the safety parameters of this strategy, our study shows no major concerns, with only four SAEs related to the plasma infusion; these findings are in line with those described in previous studies[25].

Our study has several strengths. It is the largest of its kind, studying the effect of CP for high-risk outpatients with COVID-19 early after initiation of symptoms. The fact that 93% of all patients were SARS-CoV-2 antibody negative at the time of inclusion confirms that they were recruited in the early stage of the disease. Pooling of the data from both studies was pre-planned and initiated before any interim analyses were performed and when both studies were early in their recruitment. Both teams remained fully blinded as the (interim) analyses were done by an unblinded statistical team that shared the results with the DSMB on a regular basis. The COMPILEhome study used the same primary endpoint as the CoV-Early trial, and therefore we did not perform a separate sample size calculation. As our assumptions about the outcome across the ordinal scale were somewhat different than anticipated in the original sample size calculation (fewer hospitalizations and deaths in particular), we repeated the calculation of the effect size that our study was powered for post-hoc. This showed that our study still had 80% power to detect an odds ratio of 0.65 for the primary endpoint, very close to the original power calculation. We, therefore, consider our results methodologically sound.

Several limitations should be mentioned. Although we only included patients aged ≥50, and most of them also had

comorbidities, the hospital admission and death rates were relatively low at 9.5%. Therefore, the study was not powered to exclude a small overall treatment effect on these endpoints. However, administering CP to infectious and symptomatic outpatients is complex and labor-intensive. Hence, we think that CP's clinical role is significantly diminished if unable to establish something greater than "a small effect" because it ceases to be practical. The contribution of the individual comorbidities to COVID-19 risk in our study should be interpreted cautiously because, owing to the lack of consensus regarding the relative relevance of each of them, we summed them in a non-weighted fashion. As vaccination uptake progressed in patients aged 50 or older and monoclonal antibody-based therapy with proven effectiveness in high-risk outpatients became available, the recruitment dropped dramatically as of June 2021. This resulted in the recommendation by the individual and COMPILE_home DSMBs that further enrollment was unlikely to change the results, and both studies were discontinued. Regarding the advent of the SARS-CoV-2 variants that may be less susceptible to antibodies induced by the original SARS-CoV-2 virus or the alpha variant, it is reassuring that >95% of the patients in both countries were included at a time when the delta variant was still rare (<5%) (Supplementary Figs. 3, 4). The last limitation of our study (and all studies on CP for COVID-19 so far) is the lack of a proper phase 2 dose-finding study. In a recent study, we administered 600 mL of CP to 25 SARS-CoV-2 antibody-negative B-cell depleted patients diagnosed with COVID-19[26]. While all seroconverted immediately after transfusion, the median virus neutralization titer only rose to 1:40. This is 4 times lower than the median titer in immunocompetent convalescent COVID-19 patients and up to 100 times lower than titers observed after treatment with monoclonal antibodies[7,19]. Therefore, we postulate that the range of neutralizing antibody titers present in the 200–300 mL of plasma we used may well have been too low. That underdosing may partially explain our findings is also suggested by a study in which human CP with a neutralizing antibody titer of 1:320 did not prevent disease in hamsters while a titer of 1:2560 did[27]. Hence, we recommend that any future study on CP for COVID-19 should use donors at the upper extreme end of antibody titers (e.g., >1:2560 IU). Although, this was virtually impossible in 2020, this should no longer be difficult now as plasma donors recently vaccinated or boosted with a mRNA SARS-CoV-2 vaccine can be selected.

Last but not least, a recent preprint publication by Sullivan et al. described the results of the CSSC04 study in which 1181 outpatients received one unit of convalescent or control plasma. In this trial, CP lowered the risk of hospital admission or death from 6.3 to 2.9%, $p = 0.004$[28]. Therefore, the limited impact on hospital admission or death in our study should be interpreted in the context of this trial.

In conclusion, treatment of COVID-19 with CP in the first 7 days after symptom onset did not improve the outcome. Proper dose-finding studies should be conducted, preferentially in patients with ≤5 days of symptoms before future phase 3 studies on CP are initiated.

## Methods

**Overview of study design and research partners**. Beginning in November 2020, we systematically searched for RCTs recruiting outpatients that compared treatment with CP with a blinded or unblinded control arm in the European (https://www.clinicaltrialsregister.eu/) and American (www.clinicaltrials.gov) trial register. Search terms were convalescent plasma, COVID-19, phase 2 or phase 3, adult, and recruiting or not recruiting. Studies were selected if they were RCTs on outpatients, if their inclusion criteria were confined to patients who had symptoms less than 7 days, and if they had a planned sample size of at least 100 participants of age 50 or older. Investigators of qualifying trials were contacted and informed about COMPILE_home and invited to collaborate in the study.

The full COMPILE_home protocol is available as an online supplement. The study was designed as a Bayesian-adaptive individual patient data meta-analysis of ongoing clinical trials. Prior to the start of pooling, the study teams agreed upon a minimal set of data required to analyze the primary and secondary endpoints was agreed upon by the study teams. Each trial provided updated data every 6 weeks. The pooled data were monitored by 2 unblinded statisticians and a data and safety monitoring board (DSMB) every 6 weeks using a pre-established stopping guideline for efficacy. At each interim analysis, a posterior distribution of the treatment effect was estimated.

**Study patients and selection criteria**. Although the exact inclusion and exclusion criteria could vary across the trials, all the subjects had to fulfill the following criteria; (1) Participant of a trial that joined the COMPILE_home consortium, (2) Confirmed COVID-19 diagnosis by a diagnostic PCR or antigen test of <7 days, (3) Neither hospitalized nor at the emergency room department of a hospital before or at the time of randomization, (4) Symptomatic with illness onset ≤7 days at the time of screening for the study defined by a physician with a complete clinical history, and (5) Age 50 or older. Trials had to be approved by the institutional review boards, and competent authorities of the countries involved, and all patients gave written informed consent.

**Intervention**. To qualify for COMPILE_home, participants randomly assigned to the experimental group had to receive an infusion of ABO-compatible CP with high antibody titers as determined via a semiquantitative antibody test against the spike protein or a virus neutralization assay. Only trials in which the participants were masked for the intervention were included.

**Outcomes**. Two primary efficacy outcome variables were selected. The first primary endpoint incorporated the speed of recovery as well as the risk of hospital admission, ICU admission or death in a 5-point ordinal scale. It was defined as the highest score on a 5-point ordinal disease severity scale within the 28 days after randomization. A patient scored 1 if he/she recovered quickly (i.e., fully recovered within seven days after transfusion), 2 when continued symptoms attributable to COVID-19 were present on day seven, 3 when admission to a hospital was required at any point within 28 days, 4 when invasive ventilation was required at any point within 28 days, and 5 when the patient had died at any point within 28 days. This means that the best outcome (ordinal scale score of 1) is given when a patient is fully recovered before day 8 and was never hospitalized nor died in the 28 days after transfusion, while a patient who recovered after day 7 but was never hospitalized nor died in the 28 days scored a 2 on the scale. The second primary endpoint was the occurrence of hospitalization or death within 28 days. Secondary endpoints were time to full symptom resolution (assessed by the blinded study team during a telephone contact on day 7, day 14, and day 28) and the safety of CP in outpatients with COVID-19. Pre-planned subgroup analyses assessed the efficacy of the 2 primary outcomes in the following subgroups: (1) days since disease onset (1–5 or >5 days), (2) level of neutralizing antibody anti-SARS-CoV-2 titers in transfused plasma, and (3) Negative serum anti-SARS-CoV-2 IgG status (Trimeric Spike antibody test, Liaison, Diasorin, Saluggia, Italy).

**Statistical analysis**. The first primary endpoint was analyzed with a Bayesian proportional odds model with normally distributed priors. The model included a main treatment effect shared among the trials (using a skeptical (i.e., conservative) standard deviation of 0.4), main trial effects (using standard deviation 0.5 for the prior distribution), and trial by treatment interactions (using a standard deviation of 0.14 for the prior distribution). The following covariates were included with a standard deviation of 0.5 for the prior distribution of their effects: age, sex, number of comorbidities (0–9), oxygen saturation at baseline (in %), immunocompromised state (Y/N) and duration of time (in days) since COVID-19 symptom onset (Supplementary Table 1). The second primary endpoint was analyzed with a Bayesian logistic model with a similar specification.

The use of the Bayesian framework and stopping rules enables continuous monitoring of the accrued data, and allowed for real-time decisions without penalties for multiple data looks associated with the classic frequentist approach. The results of each interim analysis were reported to the unblinded DSMB. The process and pre-specified thresholds for efficacy are described in detail in the protocol. The full statistical analysis plan is available as an online supplement.

The number of studies and patients included in COMPILE_home was not restricted and there was no pre-determined minimum or maximum sample size. The monitoring was planned to continue until the DSMB determined that there was sufficient evidence to recommend stopping the study. This situation could be achieved when the predefined stopping thresholds signaled efficacy or when the included studies had finished enrollment or any future recruitment was very unlikely to change the conclusion.

**Ethical approval**. The study was reviewed and approved by the institutional review boards of the Erasmus University Medical Center. The study was done according to the Helsinki Declaration of the World Medical Association. Written informed consent was obtained from every patient or legal representative. The COMPILE-home DSMB consisted of the chair (an infectious diseases physician), the

unblinded statisticians from the individual studies and another infectious diseases specialist. They reviewed the pooled dataset on a regular basis as described in the COMPILEhome study protocol and recommended the study team regarding the further conduct of the study. Findings are reported according to the CONSORT (Consolidated Standards of Reporting Trials) statement. For the COnV-ert study, the protocol was approved by the Ethics Committee at Hospital Germans Trias i Pujol (number PI 20-313) and the institutional review boards of participating centers. For the CoV-Early study, the protocol was approved by the medical ethical review board of the Erasmus Medical Center (METC-2020-0682).

**Reporting summary**. Further information on research design is available in the Nature Research Reporting Summary linked to this article.

## Data availability
The source data and generated data are available in the Supplementary Data files.

## Code availability
The codes generated during the current study are available from the corresponding author on reasonable request.

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

## Acknowledgements
*For the CoV-Early study group*. The DSMB members for COMPILEhome (Jan Nouwen, Andrea Troxel, Grigorios Papageorgiou, David Boulware, Josep Puig). All GGD contact tracers who informed potential study candidates. All infectiologists at Erasmus MC who worked even harder during COVID times to let the study team focus on the COVID trials (Adam Anas, Hannelore Bax, Mariana de Mendonça-Melo, Els van Nood, Jan Nouwen, Karin Schurink, Lennert Slobbe, Dorine de Vries-Sluijs) and the ER physicians and colleagues from the department of internal medicine who referred patients and facilitated patient recruitment. The medical students who contacted the patients (Romée Land, Eva Pruijt, Silje Taal, Liselotte Jeletich, Willem Sebrechts, Femke de Vries, Tia Rijlaarsdam). The many research assistants, research nurses and other participants who helped in facilitating the study (Maartje Wagemaker, Marita Tjauw Joe Kim—Amad-moestar, Diane Struik, Denise Heida-Peters, Sabine Harinck, Marjo van der Poel, Greetje van Asselt, Danielle Orij—Westerhof, Marloes Romeijn, Marlies Bouterse, Pepita de Vries, Dewi Dubbelaar, Cynthia Oud, Jo Anne den Ouden, Milly Haverkort). Stichting Hemato-Oncologie voor Volwassenen Nederland (HOVON) for building the eCRF and providing real-time support with the eCRF on very short notice (Henk Hofwegen, Ronnie van der Holt, Mirjam Stomp, Marleen Luten, Monique Steijart). Professor Eva Petkova at NYU Langone who co-designed the COMPILE concept. All the persons involved in the blinding and distribution of plasma products at the sites. Last but certainly not least, all patients that participated in the trial, the thousands of plasma donors and the COVID test centers who informed patients across the Netherlands about the study. This study was made possible by a research grant from ZONMW, the Netherlands (10430062010001). Sanquin Blood Supply provided convalescent plasma free of charge for study sites in the Netherlands. *For the COnV-ert study group*. The trial was sponsored by the Fight AIDS and Infectious Diseases Foundation with funding from the pharmaceutical company Grifols S.A. and the crowdfunding campaign YoMeCorono (www. yomecorono.com). The study received support of the Hospital Universitari Germans Trias i Pujol, and Banc de Sang i Teixits de Catalunya (BST). We thank all the plasma donors and the participants in this study and all the effort they made to attend the visits for follow-up. We thank Gerard Carot-Sans for providing medical writing support with manuscript preparation and Roser Escrig for her support in the study design and medical writing assistance with the study documentation. We also thank Laia Bertran, Mireia Clua, Jordi Mitjà, Claudia Laporte, Sergi Gavilan, Miquel Àngel Rodríguez, Joan Mercado and Enric Nieto for the operational and financial management of the project. We thank the personnel from the Fight Aids and Infectious Diseases Foundation for their support in administration, human resources and supply chain management. We thank the independent DSMB for their time and dedication: Cinta Hierro (Catalan Institute of Oncology, Badalona, Spain), Natalia Tovar (Hospital Clinic,Barcelona, Spain), Binh Ngo (University of Southern California, Los Angeles, US), David Boulware (University of Minnesota, Minneapolis, US), Robin Mogg (Bill and Melinda Gates Research Institute, Seattle, US). ISGlobal receives support from the Spanish Ministry of Science and Innovation through the "Centro de Excelencia Severo Ochoa 2019–2023" Program (CEX2018-000806-S), and support from the Generalitat de Catalunya through the CERCA Program. CISM is supported by the Government of Mozambique and the Spanish Agency for International Development (AECID). BB was supported by a Beatriu de Pinós postdoctoral fellow granted by the Government of Catalonia's Secretariat for Universities and Research, and by Marie Sklodowska-Curie Actions COFUND Program

(BP3, 801370). EP was supported by a doctoral grant from National Agency for Research and Development of Chile (ANID): 72180406. OM was supported by the European Research Council (ERC) under the European Union's Horizon 2020 research and innovation program. ZONMW, the Netherlands, grant number 10430062010001. SUP-PORT-E, grant number 101015756. YoMeCorono, www.yomecorono.com. The Fight AIDS and Infectious Diseases Foundation with funding from the pharmaceutical company Grifols S.A. The funders had no role in the design or conduct of the study; collection, management, analysis, or interpretation of the data; preparation, review, or approval of the manuscript; or decision to submit the manuscript for publication.

## Author contributions

B.J.A.R. conducted the search for studies that could potentially enter the COMPILEhome consortium. B.J.A.R. and OM agreed on the terms for the COMPILEhome consortium. B.J.A.R., O.M., P.M.M., A.G., A.A., C.R., M.C.M., B.B., A.T., and J.J.Z. conceived and designed the COnV-ert and CoV-Early studies. The COMPILEhome protocol was written by B.J.A.R. and G.P. G.P. and NvG wrote the statistical analysis plan and did the statistical analysis. P.M.M., A.G., A.A., C.R., C.G., G.P., N.v.G., C.J., G.G., F.S., E.v.d.S., M.C.M., D.O., F.P.F., P.M., S.V., V.G.G., A.R.C., A.R.M., J.R.C., R.A.S., J.R.G., J.B., I.B., J.A., Q.B., B.C., B.B., A.T., J.J.Z., O.M., B.J.A.R. acquired and interpreted the data. P.M.M., A.G., A.A., O.M., G.P., and B.J.A.R. drafted the paper. P.M.M., A.G., A.A., C.R., C.G., G.P., N.v.G., C.J., G.G., F.S., E.v.d.S., M.C.M., D.O., F.P.F., P.M., S.V., V.G.G., A.R.C., A.R.M., J.R.C., R.A.S., J.R.G., J.B., I.B., J.A., Q.B., B.C., B.B., A.T., J.J.Z., O.M., B.J.A.R. critically revised the paper for important intellectual content. O.M. and B.J.A.R. were responsible for the final decision to submit for publication. All authors have seen and approved the paper.

## Competing interests

The authors declare no competing interests.

## Additional information

[1]Fight AIDS and Infectious Diseases Foundation, Badalona, Spain. [2]ISGlobal, Hospital Clínic, Universitat de Barcelona, Barcelona, Spain. [3]Department of Internal Medicine, Section of Infectious Diseases and department of Medical Microbiology and Infectious Diseases, Erasmus MC, University Medical Center, Rotterdam, The Netherlands. [4]Hospital Universitari Germans Trias i Pujol, Badalona, Spain. [5]Facultat de Medicina-Universitat de Barcelona, Barcelona, Spain. [6]Department of Viroscience, Erasmus MC, Rotterdam, The Netherlands. [7]Department of Biostatistics, Erasmus MC, University Medical Center, Rotterdam, The Netherlands. [8]Department of Biomedical Data Sciences, Section of Medical Statistics, Leiden University Medical Center, Leiden, The Netherlands. [9]Department of Infectious Diseases and Acute Internal Medicine, Leiden University Medical Center, Leiden, The Netherlands. [10]Unit of Transfusion Medicine, Sanquin Blood Supply, Amsterdam, The Netherlands. [11]Department of Experimental Immunohematology, Sanquin Research, Amsterdam, The Netherlands. [12]Hospital Universitari Parc Taulí I3PT, Sabadell, Spain. [13]Universitat Autònoma de Barcelona, Barcelona, Spain. [14]Bioclever-CRO, Barcelona, Spain. [15]Emergency Department, Bellvitge University Hospital, L'Hospitalet de LLobregat, Barcelona, Spain. [16]Clinical Research Support Unit (HUB-IDIBELL: Bellvitge University Hospital & Bellvitge Biomedical Research Institute), Bellvitge University Hospital, L'Hospitalet de Llobregat, Barcelona, Spain. [17]Pharmacology Unit, Department of Pathology and Experimental Therapeutics, School of Medicine and 33 Health Sciences, IDIBELL, University of Barcelona, L'Hospitalet de Llobregat, Barcelona, Spain. [18]Unitat de Suport a la Recerca de la Catalunya Central, Fundació Institut Universitari per a la recerca a l'Atenció Primària de Salut Jordi Gol i Gurina, Sant Fruitós de Bages, Spain. [19]Health Promotion in Rural Areas Research Group, Gerència Territorial de la Catalunya Central, Institut Català de la Salut, Sant Fruitós de Bages, Spain. [20]Universitat de Vic—Universitat Central de Catalunya (UVIC-UCC), Vic, Spain. [21]Salut Catalunya Central, Hospital de Berga, Berga, Spain. [22]Blood Bank Department—Banc de Sang i Teixits (BST), Barcelona, Spain. [23]IrsiCaixa AIDS Research Institute, Germans Trias i Pujol Research Institute (IGTP), Badalona, Spain. [24]Metropolitana Nord Laboratory, Institut Català de la Salut, Badalona, Spain. [25]Gerència Territorial Metropolitana Nord, Institut Català de la Salut, Barcelona, Spain. [26]Centro de Investigação em Saúde de Manhiça (CISM), Maputo, Mozambique. [27]Pg. Lluís Companys 23, ICREA, Barcelona, Spain. [28]Pediatrics Department, Hospital Sant Joan de Déu, Universitat de Barcelona, Esplugues, Spain. [29]Consorcio de Investigación Biomédica en Red de Epidemiología y Salud Pública (CIBERESP), Madrid, Spain. [30]Department of Population Health, NYU Grossman School of Medicine, New York, NY, USA. [31]Department of Haematology, Leiden University Medical Centre, Leiden, The Netherlands. [32]CCTR, Sanquin Blood Supply, Amsterdam, The Netherlands. [33]Lihir Medical Centre—InternationalSOS, Lihir Island, Papua New Guinea. [62]These authors contributed equally: Pere Millat-Martinez, Arvind Gharbharan, Andrea Alemany. [63]These authors jointly supervised this work: Jaap Jan Zwaginga, Oriol Mitjà, Bart J.A. Rijnders. *Lists of authors and their affiliations appear at the end of the paper. ✉email: b.rijnders@erasmusmc.nl

## CoV-Early study group

Arvind Gharbharan[34], Casper Rokx[34], Carlijn Jordans[34], Corine Geurtsvankessel[34], Grigorios Papageorgiou[34], Bart Rijnders[34], Peter Katsikis[34], Yvonne Müller[34], Marion Koopmans[34], Susanne Bogers[34], Jelle Miedema[34], Henk Russcher[34], Cees Scherpenisse[34], Rene van Engen[34], Ayten Karisli[34], Hannelore Götz[35], Jelle Struik[36], Lotte Rokx-Niemantsverdriet[36], Nan van Geloven[37], Geert Groeneveld[37], Jaap Jan Zwaginga[37], Lisa Zwaginga[37], Josine Oud[37], Romy Meier[37], Erik van Zwet[37], Simon Mooijaart[37], Arjan Albersen[37], Francis Swaneveld[38],

Ellen van der Schoot[38], Hans Vrielink[38], Leo van de Watering[38], Boris Hogema[38], Peter van Wijngaarden[39], Ronald van Etten[39], Adriaan van Gammeren[39], Nanda Maas[39], Betty van Ginneken[39], Jan den Hollander[40], Jose Verstijnen[40], Juliette van den Berg – Rahman[40], Faiz Karim[41], Siepke Hiddema[41], Kim van Elst[41], Elena van Leeuwen-Segarceanu[42], Annette Reitsma[42], Karin Molenkamp[42], Robert Soetekouw[43], Caterina Band[43], José de Droog[43], Jolanda Lammers[44], Lonneke Buitenhuis[44], Douwe Postma[45], David Koster[45], Michaèl Lukens[45], Thea Scholtens[45], Maartje van den Boomgaard[45], Machiel Vonk[45], Linda Kampschreur[46], Marit van Vonderen[46], Loes Vrolijk[46], Chantal Reusken[47], Johan Reimerink[47] & Heli Harvala[48]

[34]Erasmus MC, Rotterdam, The Netherlands. [35]GGD Rotterdam, Rotterdam, The Netherlands. [36]Gezondheidscentrum Mathenesserlaan, Rotterdam, The Netherlands. [37]Leids Universitair Medisch Centrum, Leiden, The Netherlands. [38]Sanquin Blood Supply, Amsterdam, The Netherlands. [39]Amphia Hospital, Breda, The Netherlands. [40]Maasstad ziekenhuis, Rotterdam, The Netherlands. [41]Groene hart hospital, Gouda, The Netherlands. [42]St Antonius hospital, Nieuwegein, The Netherlands. [43]Spaarne Gasthuis, Hoofddorp, The Netherlands. [44]Isala Hospital, Zwolle, The Netherlands. [45]Universitair medisch centrum Groningen, Groningen, The Netherlands. [46]Medical Center Leeuwarden, Leeuwarden, The Netherlands. [47]Centrum voor Infectieziektebestrijding, RIVM, Bilthoven, The Netherlands. [48]NHS Blood and Transplant, London, UK.

## COnV-ert study group

Andrea Alemany[49], Marc Corbacho-Monné[49], Dan Ouchi[49], Bonaventura Clotet[49], Oriol Mitjà[49], Gèlia Costes[49], Mar Capdevila-Jáuregui[49], Pamela Torrano-Soler[49], Alba San José[49], Zahida Jiménez[49], Ferran Ramírez-Viaplana[49], Susana Ferrer[49], Mireia Gallardo[49], Maria Ubals[49], Camila González-Beiras[49], Martí Vall-Mayans[49], Miquel Angel Rodriguez-Arias[49], Clara Suñer[49], Jordi Puig[49], Aroa Nieto[49], Ivan Galvan-Femenia[49], Xavier Comas-Leon[49], Pere Millat-Martínez[50], Quique Bassat[50], Bàrbara Baro[50], Ignacio Blanco[51], Jordi Ara[51], Glòria Bonet Papell[51], Maria Delgado Capel[51], Beatriz Díez Sánchez[51], Maria Pons Barber[51], Cristian Gonzalez Ruiz[51], Laura Navarrete Gonzalez[51], David González García[51], Ainhoa Vivero Larraza[51], Victor Carceles Peiró[51], Clàudia Roquer López[51], Magí Ferrer[51], Pierre Malchair[52], Sebastian Videla[52], Vanesa García García[52], Carlota Gudiol[52], Aurema Otero[52], Jose Carlos Ruibal Suarez[52], Alvaro Zarauza Pellejero[52], Ferran Llopis Roca[52], Orlando Rodriguez Cortez[52], Pablo Casares Gonzalez[52], Gemma Arcos Vila[52], Begoña Flores Aguilera[52], Graciela Rodríguez-Sevilla[52], Macarena Dastis Arias[52], Anna Ruiz-Comellas[53], Anna Ramírez-Morros[53], Judit Roca Font[53], Katherine M. Carrasco Matos[53], Glòria Saüch Valmaña[53], Carla Vidal Obradors[53], Joana Rodríguez Codina[54], Rosa Amado Simon[54], Silvia Tarres García[54], Margarida Curriu Sabatès[54], Raquel Nieto Rodríguez[54], Joan-Ramon Grífols[55], Anna Millan[55], Enric Contreras[55], Àgueda Ancochea[55], Rosa Línio[55], Miriam Fornos[55], Natàlia Casamitjana[55], Eva Alonso[55], Núria Martinez[55], Laura Analía Maglio[55], Laura Comellas Fernandez[55], Nadia Garcia[55], Luis Hernández[55], María Isabel González[55], Anna Bravo[55], Yolanda García[55], Núria Prat[56], Joaquim Verdaguer[56], Thatiana Vértiz Guidotti[56], Sergio Benavent[56], Andrea Sofia Bianco[56], Ney Nicanor Briones Zambrano[56], Maria Viozquez Meya[56], Anna Forcada[57], Josep Vidal-Alaball[57], Montserrat Giménez[58], Alexa París[58], Gema Fernández Rivas[58], Cristina Casañ Lopez[58], Águeda Hernández[58], Antoni E. Bordoy[58], Victoria González Soler[58], Julian Blanco[59], Edwards Pradenas[59], Silvia Marfil[59], Benjamin Trinité[59], Francini Piccolo Ferreira[60], Mireia Bonet[60], Jordi Cantoni[60] & Michael Marks[61]

[49]Fight AIDS and Infectious Diseases Foundation, Badalona, Spain. [50]Barcelona Institute for Global Health, ISGlobal, Barcelona, Spain. [51]Hospital Universitari Germans Trias i Pujol (HUGTiP), Badalona, Spain. [52]Hospital Universitari de Bellvitge, L'Hospitalet de Llobregat, Barcelona, Spain. [53]CUAP Manresa, Manresa, Spain. [54]Hospital Comarcal de Sant Bernabé, Berga, Spain. [55]Blood Bank Department—Banc de Sang i Teixits (BST), Barcelona, Spain. [56]Gerència Territorial Metropolitana Nord, Badalona, Spain. [57]Gerència Territorial Catalunya Central, Sant Fruitós de Bages, Spain. [58]Metropolitana Nord Laboratory, Badalona, Spain. [59]IrsiCaixa AIDS Research Institute, Badalona, Spain. [60]Bioclever-CRO, Barcelona, Spain. [61]London School of Hygiene and Tropical Medicine, London, UK.

