## [Peer Review File · Nature Communications]

Prospective individual patient data meta-analysis of two randomized trials on convalescent plasma for COVID-19 outpatientsReviewers' Comments:

Reviewer #1:

Remarks to the Author:

Peer review: NCOMMS-21-46656 (Millat-Martinez P et al.)

Summary: Thank you for the opportunity to review this manuscript. The authors present a prospective individual patient data meta-analysis of two randomised controlled trials of convalescent plasma administered within 7 days of symptom onset in COVID-19 patients, who did not require (prior to) hospitalisation at the time of randomisation. This IPDMA reports data from 797 patients (390 randomised to CP and 392 to placebo). The pre-defined primary outcome was not different between the two groups.

Major concerns

1. Neither trial completed their pre-planned sample size.
2. The authors report a post-hoc (not-prespecified cut off of ≤ 5 or > 5 days).
3. There are at least two major trials that are not part of the IPDMA, that evaluated similar question. Please see - <https://www.medrxiv.org/content/10.1101/2021.12.10.21267485v1> and <https://www.ncbi.nlm.nih.gov/pubmed/34407339>
4. Whilst IPDMA may be an important methodological plus, lack of key published trial data makes this whole exercise of limited utility.
5. The above issues are compounded by variants, monoclonal antibodies, antivirals, and potential impact of vaccination in outpatient settings.
6. There is a Cochrane meta-analysis, WHO summary and numerous other meta-analysis including data in similar patient population.

In summary, the timeliness and utility of this work to advance the field is limited.

Reviewer #2:

Remarks to the Author:

Thanks for the opportunity to review this manuscript. I believe this is an interesting and well-designed piece of work.

Among its strengths are the individual patient pooled data analysis, the early implementation of the merging strategy between the ongoing studies, the selected outcomes and the statistical analysis plan.

There are few remaining questions for the authors:

-As adequately commented in the manuscript, the timeliness of the intervention with any passive immunity strategy is key. As such, was any bias-avoiding strategy implemented for defining the moment of symptom onset? Was it defined exclusively through patient self-report? Was there any maximum time allowed between the positive SARS-CoV-2 test and the date of initiation of symptoms?

-Was there a minimum antibody threshold established for the convalescent plasma units that were infused?

-Comorbidities were operationalized as cumulative number. As such, all comorbidities were equally weighted. Although it appears clear that some comorbidities were associated with increased risk of severe disease.

-Have you considered including post-hoc sensitivity analysis based on baseline antibody status and SARS-CoV-2 vaccination status?

Thanks for considering these comments.

Reviewer #3:

Remarks to the Author:

Overall, this is an interesting paper. In a sense, it is a preemptive meta analysis but unfortunately lacks the systematic review component. The two partnering clinical trials each had their own challenges in recruitment and while the data are combined, it is not clear the study was powered or should be concluded as conservatively as presented.

The primary outcome measure is not fully objectively defined. The most ideal state – recovered – appears to be defined within 7 days. The remaining events are based on 28 days of following (presumably with no attrition or loss to follow up). The “recovered” is actually “recovered quickly” as many of the other categories would also have resulted in a recovered state. This should be clarified and support for recovered within 7 days needs to be justified.

In a similar manner, the time zero is the date of the transfusion. This is reasonable, but showing some tables and or figures based on time of either positive test and /or time of suspected infection would be helpful.

Table 1 – Why such limited data? Presumably additional data was gathered and pooled. A more thorough description of the sample should be provided. At a minimum, a frequency distribution of the comorbidities would be welcomed.

The data in Table 2 is a little hard to interpret because the ordinal nature of the data is lost without the cumulative percentages being presented. Ideally, the data would be plotted to show the ordinal relationship and the pairing between treatment groups using a variation of a stacked barchart (e.g., <https://www.nejm.org/doi/full/10.1056/nejm199512143332401>).

The data in figure 2 is unexpected. What is “full symptom resolution”? Table 2 would suggest this would be >90% for both groups. This figure is not consistent with that. Presumably there is informative censoring that is affecting the results. This warrants further consideration.

The supplemental results show a strong difference related to study. This was not presented in any detail in the main body. Given I have concerns about selection bias in the included pooled results and a strong difference by study, there are concerns if these data are interpretable in the way the authors present. At a minimum, study-specific estimates need to be given throughout the paper as they are key to the understanding of the data. Essentially, this is neither a systematic review or a comprehensive individual patient level meta analysis.

The conclusions are very conservative and written in a “frequentist” confirming the null hypothesis manner. This is unexpected given the methods suggest this is a Bayesian analysis. Objectively speaking, there is a promising “signal” for all outcomes, particularly for the true early treated. What this study appears to be is essentially a non-conclusive study largely due to insufficient sample size.

Speaking of sample size, the sample size on pdf page 53 OF THE CONV-ERT in the supplement expected a 50% reduction in hospitalization. This was a large difference, potentially greater than could be expected. However, what is more interesting on this page is the discussion of the viral load. This data would be the mechanistic data for treatment. Why was this data not presented?

Reviewer #4:

Remarks to the Author:

It's important for studies of convalescent plasma to be reported so that in aggregate they can inform the medical community about how efficacy might differ by timing, quality of CP, patient characteristics etc. The challenges of doing randomized double blind CP trials were formidable and it's a credit that these trials were launched and completed. Pooling ongoing trials as a practical remedy to counteract slow recruitment in individual is a great idea and important to circulate.

Evaluating the treatment every 6 weeks with no adjustment for multiple looks and no planned end is natural from the Bayesian viewpoint but makes it difficult to understand what the strength of evidence really is, especially so for a medical audience. Since this paper reports a null result, this concern is not critical, but had the study showed a benefit, it would be important to understand the frequentist operating characteristics of this approach. Some complexities include the 2 related guidelines for stopping (one involving the PO and logistic model, the other the PO model) and the monitoring every 6 weeks which could end up being many looks between few patients for a long trial with slow accrual. Such a cadence could meaningfully increase the chance of a false positive.

The study stopped because of dwindling accrual and a changing COVID landscape. It would be good to report the planned sample size for each study to get a feel for the fraction of planned information that was achieved. It is a little complicated though because you did pool the two trials so if each trial achieved 50% of the planned sample size, maybe that would be close to the planned information.

Another approach would be to do a post-hoc power calculation that used the lumped data to estimate nuisance parameters and then report what treatment effect could be detected with posterior probability of 80% or 90% based on a single analysis of the entire dataset.

You report skeptical standard deviations of 0.4, 0.5 or 0.14. Is there a way to better convey what that means to a medical audience?

In the abstract could say the endpoint is death or hospitalization by 28 days.

You write the standard deviation for the covariates was 0.5, really that's the standard deviation for the regression coefficient.

For the log-rank test, how were the deaths treated? Presumably censored at day 29. However, if a patient enrolled on June 5 died on June 10 and the data was frozen on June 15, then that patient should be censored at day 10 post-randomization, i.e. the Fine-Grey approach. I suspect you have no such patients so censoring all deaths at day 29 would be fine.

I'm not sure that your pseudo-code corresponds to a Dirichlet distribution with a one vector for the parameters.

Reviewer #1 (Remarks to the Author):

Peer review: NCOMMS-21-46656 (Millat-Martinez P et al.)

Summary: Thank you for the opportunity to review this manuscript. The authors present a prospective individual patient data meta-analysis of two randomised controlled trials of convalescent plasma administered within 7 days of symptom onset in COVID-19 patients, who did not require (prior to) hospitalisation at the time of randomisation. This IPDMA reports data from 797 patients (390 randomised to CP and 392 to placebo). The pre-defined primary outcome was not different between the two groups.

Major concerns

1. Neither trial completed their pre-planned sample size.

Reply:

Thanks for taking time in reviewing our manuscript. We understand the reviewer's concern regarding the individual trials not completing their pre-planned sample size. Like the reviewer pointed, we designed a prospective individual patient meta-analysis to be able to achieve a higher sample size. Precisely, we believe that this is one of the strong points of the COMPILEhome pooled analysis and one of the main reasons for conducting this study, as we were able to achieve a sample size of 797 participants by pooling the data of ongoing studies in real-time; this sample size is higher than in any of the studies published assessing CP for outpatients so far. The COMPILEhome had a Bayesian analysis methodology where there are stopping rules for effectivity with multiple interim analyses as described in the statistical analysis plan.

2. The authors report a post-hoc (not-prespecified cut off of ≤ 5 or >5 days).

Reply:

We would like to explain that this was a pre-defined secondary endpoint; as described in the methods "*Pre-planned subgroup analyses assessed the efficacy of the 2 primary outcomes in the following subgroups: 1) days since disease onset (1-5 or >5days), [...]*" as well as in the COMPILEhome SAP (p.12).

3. There are at least two major trials that are not part of the IPDMA, that evaluated similar question. Please see - <https://www.medrxiv.org/content/10.1101/2021.12.10.21267485v1> and <https://www.ncbi.nlm.nih.gov/pubmed/34407339>

Reply:

We did not include these two trials for different reasons. First, the Korley et al. paper recruited patients visiting the emergency room which is a very different population. We focused on patients outside the clinic, so early after the start of symptoms and without the need of supplemental oxygen. Indeed, As described in the methods of COMPILEhome protocol, one of the criteria to contact research teams to ask them to join COMPILEhome was: "*Neither hospitalized nor at the emergency room department of a hospital before or at the time of randomization*". That this is a different population is nicely illustrated by the fact that many patients that they enrolled at the ER

eventually could not be discharged and were admitted to the hospital directly on the day of randomization.

The second trial by David Sullivan et al. is indeed very relevant and could fulfill our inclusion criteria. We have been in contact several times with dr. David Sullivan and his team. They decided to continue their trial without participating in COMPILEhome. In fact, we refer to their study in our results as follows: *“Of the four remaining studies, one study team opted to abstain from pooling data while another never responded to repeated emails and calls, resulting in two trials included in the pooled analysis: The CONV-ert study (NCT04621123) and the CoV-Early study (NCT04589949)”*. When we submitted our paper, the results of the study by Sullivan et al were unpublished. As they are now available as a preprint, we address their study results in the discussion of our paper as follows: *“Last but not least, a recent preprint publication by Sullivan et al. described the results of the CSSC04 study in which 1181 outpatients received one unit of convalescent or control plasma. In this trial, ConvP lowered the risk of hospital admission or death from 6.3 to 2.9%, $p=0.004$ (26). Therefore, the limited impact on hospital admission or death in our study should be interpreted in the context of this trial.”*

We certainly have the intention, and we are already in contact with dr. David Sullivan to perform a retrospective individual patient data meta-analysis of the CSSC4, Cov-Early, CON-Vert and if possible, also Libster et al trial.

4. Whilst IPDMA may be an important methodological plus, lack of key published trial data makes this whole exercise of limited utility.

Reply:

We value the reviewer perspective; however, we think the results of the COMPILEhome study are important precisely because of the sample size, which it is larger than all other trials on ConvP for outpatients with COVID19 except for the study by Sullivan et al. as discussed above. As explained above our study is not a retrospective IPDMA. We have put all possible effort into convincing as many as possible trials to prospectively (i.e. before their trial results were known) join our initiative. The important difference between a typical retrospective IPDMA and a prospectively designed meta-analysis is discussed in this BMJ paper by Anna Lene Seidler et al.

(<https://www.bmj.com/content/367/bmj.l5342>). The Bayesian analysis methodology of our study with multiple interim analyses and stopping rules for effectivity takes this prospective IDPMA one step further. For a statistical discussion of the methodology, we would like to refer to this recent paper by one of our co-authors <https://www.ncbi.nlm.nih.gov/pmc/articles/PMC8441650/pdf/SIM-9999-0.pdf>

We would like to stress that this is the first publication of the key results from two pivotal trials. The results of the individual trials (with their predefined primary and secondary endpoints) will be published separately. We have put all possible effort in getting the COMPILEhome analysis and paper submitted first to generate data that we need as clinicians as soon as possible and given the current situation of CP in the therapeutic landscape for COVID-19 (also discussed in our answer to the next question), we strongly believe that our results are clinically relevant and necessary to disseminate.

5. The above issues are compounded by variants, monoclonal antibodies, antivirals, and potential impact of vaccination in outpatient settings.

Reply:

We value the reviewer's perspective. However, in most parts of the world, monoclonals nor antiviral agents are available or will become available shortly. Studies on convalescent plasma for outpatients with COVID-19 are therefore still relevant.

Regarding the monoclonal antibodies' therapies, except for Sotrovimab, all available monoclonals lost activity against Omicron.

We do agree that the interpretation of the results of our study in the context of vaccinated patients is not straightforward. But this applies to all other treatments as well as they have all been studied in unvaccinated patients. Also, in the future, vaccines might not cover some emerging variants of concern, while CP could be used with virus neutralizing activity against the (new) variant that is being treated. This issue is explicitly discussed in the discussion section.

6. There is a Cochrane meta-analysis, WHO summary and numerous other meta-analysis including data in similar patient population.

Reply:

On top of what we discuss in our answer to question 4 we would like to point out that the Cochrane meta-analysis, the RECOVERY trial and other published meta-analysis on ConvP mainly include hospitalized patients. Although we acknowledge in the Introduction section this body of evidence, the population in our study consists of **unhospitalized** patients and excludes patients presenting at the emergency room of hospitals. At the moment of the submission of this paper, the only studies we were aware that tested CP for Covid19 outpatients are the study by Libster et al (treatment of elderly outpatients within 72h of symptom onset) and Korley et al (patients at the emergency room which we explicitly excluded from COMPILEhome). At this moment, and as explained above, the results of the study by Sullivan et al are now available as a preprint and we have added a paragraph in the discussion section regarding the results of this study.

7. In summary, the timeliness and utility of this work to advance the field is limited.

Reply:

We believe that our results are important for the scientific community, in particular in the context of variants that are no longer susceptible to most of the available monoclonal antibodies, and in a setting where the availability of new therapies is limited to a small part of the world.

Moving one step forward, in the Covid pandemic we were hopeful regarding CP because of previous experiences in other epidemics and it is expected that this therapeutic strategy will arise again in future outbreaks of new (viral) agents. Our results suggest that previous evidence on CP was incomplete or weak; hence, our findings settle an important basis for future attempts of using CP. Moreover, the results from our study become even more relevant now that the positive results of the trial by D. Sullivan et al have become available. Indeed, both positive and negative trials together tell the whole story.

Reviewer #2 (Remarks to the Author):

Thanks for the opportunity to review this manuscript. I believe this is an interesting and well-designed piece of work.

Among its strengths are the individual patient pooled data analysis, the early implementation of the merging strategy between the ongoing studies, the selected outcomes and the statistical analysis plan.

There are few remaining questions for the authors:

-As adequately commented in the manuscript, the timeliness of the intervention with any passive immunity strategy is key. As such, was any bias-avoiding strategy implemented for defining the moment of symptom onset? Was it defined exclusively through patient self-report? Was there any maximum time allowed between the positive SARS-CoV-2 test and the date of initiation of symptoms?

Reply:

We thank the reviewer for taking time in the revision of our manuscript. In both studies, a maximum time from symptom onset to inclusion was 7 days and both protocols excluded patients with >7days between the first positive test and inclusion. The first day of symptoms was indeed defined after a detailed patient history; although we agree that this may be somehow subjective because the very first symptoms may be mild, we think this is the only practically useful way to define symptom onset. In retrospect, this strategy seemed to have worked well. Indeed, as described in the results, the way we were able to document that almost all patients were very early in their disease course is by testing for the presence of antibodies at the time of transfusion, which typically start to be produced around day 7: >90% of all patients enrolled were still antibody negative at the time of plasma transfusion!

We have now clarified this better in the Methods section: *“Although the exact inclusion and exclusion criteria could vary across the trials, all the subjects had to fulfill the following criteria; 1) Participant of a trial that joined the COMPILEhome consortium, 2) Confirmed COVID-19 diagnosis by a diagnostic PCR or antigen test of <7 days, 3) Neither hospitalized nor at the emergency room department of a hospital before or at the time of randomization, 4) Symptomatic with illness onset ≤7 days at the time of screening for the study defined by a physician with a complete clinical history, and 5) Age 50 or older.”*

-Was there a minimum antibody threshold established for the convalescent plasma units that were infused?

Reply:

In both protocols, this was indeed strictly defined. See the supplemental data file on page 13-14 in detail and the individual study protocols (annex 1 of CONV-ERT protocol in which the Euroimmune test was used as an antibody threshold, and paragraph 6.1 of the Cov-Early protocol in which a virus neutralization test was used). On top of referring to the details in the suppl data, we have added this summary of the methodology in the main paper as well; *“The CONv-ert study selected the convalescent plasma after being screened for high anti-SARS-CoV-2 IgG titers with ELISA*

(EUROIMMUN ratio ≥ 6), according to guidelines, and supplied by the regional blood bank (Banc de Sang i Teixits de Catalunya – BST); and the CoV-Early study selected the convalescent plasma based on a plaque reduction neutralization test (PRNT) 50 titer of 1:160 or higher”

-Comorbidities were operationalized as cumulative number. As such, all comorbidities were equally weighted Although it appears clear that some comorbidities wer associated with increased risk of severe disease.

Reply:

Thank you for raising this concern. We agree that some comorbidities may be more important than others, but we do not think that sufficient consensus exists to give more weight to one comorbidity over the other. Also, we anticipated that including each comorbidity separately would lead to sparse data. We have added the following sentence in the Discussion section: *“The contribution of the individual comorbidities to COVID-19 risk in our study should be interpreted cautiously because, owing to the lack of consensus regarding the relative relevance of each of them, we summed them in a non-weighted fashion.”*

-Have you considered including post-hoc seneitivity analysys based on baseline antibody status and SARS-CoV-2 vaccination status?

Reply:

This was actually a predefined endpoint and the results were already included in the paper and described as follows: *“Also, no notable difference was observed when patients with IgG anti-SARS-CoV-2 antibodies detected at baseline were excluded (OR 0.880 95% CI 0.590-1.310 for the binary outcome, 0.892 95% CI 0.643-1.236 for ordinal outcome, Appendix Figures 7 and 8). “*

Reviewer #3 (Remarks to the Author):

1. Overall, this is an interesting paper. In a sense, it is a preemptive meta analysis but unfortunately lacks the systematic review component. The two partnering clinical trials each had their own challenges in recruitment and while the data are combined, it is not clear the study was powered or should be concluded as conservatively as presented.

Reply:

Thanks for taking the time to review our paper. In this prospective individual patient meta-analysis, we merged 2 trials at a time when they had enrolled a limited number of patients. This design is exceptional because it has allowed us to increase the power that both studies had separately. This approach is indeed without many precedents (as far as we know only 1 that has recently been accepted at JAMA open network in which the same approach was taken for studies on COVID in hospitalized patients). We have been able to obtain a larger sample size with a lower risk of a type 2 error, shorten the time to result and avoid that vaccination uptake in each of the countries (which lowers the risk of hospital admission which is part of the primary endpoint) would result in 2 individual trials that are underpowered.

We would also like to refer to our answers to question 4 and 6 of reviewer 1 in which we discuss the prospective design of our meta-analysis.

Regarding the question about statistical power: We agree that we did not explicitly mention simulations about the power and sample size requirements of the study. However, because the primary endpoint of the COMPILEhome study is identical to the endpoint of the Cov-Early study, the power and sample size estimates as described on page 35 of the Cov-Early protocol apply to the COMPILEhome study as well: *“Assuming a common odds ratio of 0.63, using a two-sided test in a proportional odds model, assuming 15% attrition (loss to follow up and admission events occurring before inclusion) a sample size of 690 is needed for a study with a power of 80% (923 for 90% power)”*.

That being said, the assumptions about the distribution across the 5 ordinal scale categories turned out to be somewhat different than anticipated (fewer hospitalizations and deaths in particular). Therefore, we additionally assessed the effect size we were actually powered for taking into account the actual distribution across the 5 categories that we observed in the control arm, as requested by the statistical reviewer. With the observed distribution over the 5 categories, the effect size that was detectable with 80% power given the 782 evaluable patients turned out very similar and actually slightly smaller than the planned effect size in the CovEarly trial (OR 0.65 was already detectable instead of OR of 0.63 with 80% power).

We therefore added this in the discussion: *“As our assumptions about the outcome across the ordinal scale were somewhat different than anticipated in the original sample size calculation (fewer hospitalizations and deaths in particular), we repeated the calculation of the effect size that our study was powered for post-hoc. This showed that our study still had 80% power to detect an odds ratio of 0.65 for the primary endpoint, very close to the original power calculation”*.

2. The primary outcome measure is not fully objectively defined. The most ideal state – recovered – appears to be defined within 7 days. The remaining events are based on 28 days of following (presumably with no attrition or loss to follow up). The “recovered” is actually “recovered quickly” as many of the other categories would also have resulted in a recovered state. This should be clarified and support for recovered within 7 days needs to be justified.

Reply:

For all patients, the first day that they were without symptoms was registered. Also, for all patients up until day 28, hospital admission status, ICU admission status and mortality was registered. With these data the ordinal scale was made. It not only reflects serious disease (hospital admission or worst) but also the speed of recovery. As described in the COMPILEhome protocol (suppl data file) the score is defined on day 28 as: 1=recovered on d7 (day 7 after transfusion, which indeed is “recovered quickly” as the reviewer pointed); 2=not recovered on d7; 3=hospital admission; 4=ICU admission for invasive ventilation; 5=death.

To help the reader with the interpretation of the primary outcome measure we have modified this paragraph and added the following information in the method section: *“The first primary endpoint incorporated the speed of recovery as well as the risk of hospital admission, ICU admission or death in a 5-point ordinal scale. It was defined as the highest score on a 5-point ordinal disease severity scale within the 28 days after randomization. A patient scored 1 if he/she recovered quickly (i.e., fully*

recovered within seven days after transfusion), 2 when continued symptoms attributable to COVID-19 were present on day seven, 3 when admission to a hospital was required at any point within 28 days, 4 when invasive ventilation was required at any point within 28 days, and 5 when the patient had died at any point within 28 days. This means that the best outcome (ordinal scale score of 1) is given when a patient is fully recovered before day 8 and was never hospitalized nor died in the 28 days after transfusion, while a patient who recovered after day 7 but was never hospitalized nor died in the 28 days scored a 2 on the scale. “

3. In a similar manner, the time zero is the date of the transfusion. This is reasonable, but showing some tables and or figures based on time of either positive test and /or time of suspected infection would be helpful.

Reply:

Given the randomized prospective design, patients were followed from randomization (which was always on the same day of the ConvP transfusion) onwards. We did not collect events in the period before that. Using time of positive test or day 1 of symptoms as time zero would mean we include periods where we cannot compare groups yet as randomization had not yet taken place and would lead to an ‘immortal time period’ between time zero and day of randomization/inclusion. For this reason, we chose to only report on outcomes using randomization/transfusion as time zero.

4. Table 1 – Why such limited data? Presumably additional data was gathered and pooled. A more thorough description of the sample should be provided. At a minimum, a frequency distribution of the comorbidities would be welcomed.

Reply:

We agree and have now provided frequency distribution on comorbidities and on age as well in the table.

5. The data in Table 2 is a little hard to interpret because the ordinal nature of the data is lost without the cumulative percentages being presented. Ideally, the data would be plotted to show the ordinal relationship and the pairing between treatment groups using a variation of a stacked barchart (e.g., <https://www.nejm.org/doi/full/10.1056/nejm199512143332401>).

Reply:

We hope that our answer to question 2 of this reviewer helps to better understand table 2. We also added some extra words in table 2 to improve its readability and include a barchart figure as suggested.

6. The data in figure 2 is unexpected. What is “full symptom resolution”? Table 2 would suggest this would be >90% for both groups. This figure is not consistent with that. Presumably there is informative censoring that is affecting the results. This warrants further consideration.

Reply:

Figure 2 gives the time to full symptom resolution. Indeed, all patients were contacted on several occasions after inclusion (day 7, 14 and again day 28) and asked if they still had any symptoms that had not been present preceding their COVID19 infection and if symptoms were no longer present what the date of the first day without symptoms was. As expected, not all patients had recovered completely on day 28 which is illustrated by figure 2. This information is not available in table 2. In table 2 it is only reported what the percentage of recovered patients was before day 8 (17.6% overall, which is in line with the level of the curves in Figure 2 at day 7). Recovery percentages up to day 28 cannot be derived from Table 2.

7. The supplemental results show a strong difference related to study. This was not presented in any detail in the main body. Given I have concerns about selection bias in the included pooled results and a strong difference by study, there are concerns if these data are interpretable in the way the authors present. At a minimum, study-specific estimates need to be given throughout the paper as they are key to the understanding of the data. Essentially, this is neither a systematic review or a comprehensive individual patient level meta analysis.

Thank you for raising these concerns. First, we would like to point out that our study is indeed not a standard retrospective individual patient data meta-analysis, it is a prospective IPD meta-analysis of two highly similar studies. This means that inclusion of trials into our study, generation of hypotheses and writing of analysis plan were all performed before the results of the individual trials were unblinded. This design does not allow any selection bias. Also, 100% of the patients enrolled and transfused in both trials were included in the COMPILEhome analysis (see consort flow diagram fig1 for details). So, there is no selection on individual patient level either. We refer to our answer to question 4 of reviewer 1 for further explanation about our design.

It is correct that patients in Spain had an overall higher score on the ordinal scale and so a poorer outcome. This resulted in the OR of 2.6 for patients included in Spain. We can only speculate why this is because this increased risk for Spanish patients was independent of age, sex and number of comorbidities (as also illustrated in the same figure). However, the treatment effect of convalescent plasma in both trials was not very different. This is illustrated by the first 2 odds ratios in appendix figure 5, which both cross 1 and are both close to 1. So while patients in Spain did worse, the effect of convalescent plasma on the primary endpoint was not very different between trials. We believe this is an important concern and worth to be mentioned in the main manuscript, so we have added the following text in the Results section *“Although being included in the CONV-ert trial was associated with a poorer overall outcome, the effect size of ConvP in each of the trials was not. This increased risk for patients in CONV-ert was independent of age, sex, and the number of comorbidities (figure S5).”*

8. The conclusions are very conservative and written in a “frequentist” confirming the null hypothesis manner. This is unexpected given the methods suggest this is a Bayesian analysis. Objectively speaking, there is a promising “signal” for all outcomes, particularly for the true early treated. What this study appears to be is essentially a non-conclusive study largely due to insufficient sample size.

Reply:

We do not agree that we had insufficient sample size for the primary endpoint. Please see our reply to the statistical reviewer (Reviewer number 4) regarding power for the primary endpoint. We

believe we correctly interpreted our Bayesian results as a negative result for the total study group (OR for primary endpoint 0.936 (posterior mean, 95% credible interval 0.667-1.311) with only a 64.9% posterior probability of benefit whereas we aimed for an OR of about 0.65).

We do agree that we see a promising 'signal' regarding patients treated 5 or fewer days after start symptoms. And we do agree that the power of our study for the second primary endpoint of hospital admission or death is more limited. We now address this in the discussion within the context of the positive findings described by D. Sullivan et al. which have become available as a preprint recently. In this study a significant reduction in hospital admission was observed. When we asked dr. D Sullivan et al. about the effect in the patients with 5 or fewer days of symptoms they confirmed that the effect size was by far the largest in this subgroup in their study as well.

This paragraph was added in the discussion; *"Last but not least, a recent preprint publication by Sullivan et al. described the results of the CSSCO4 study in which 1181 outpatients received one unit of convalescent or control plasma. In this trial, ConvP lowered the risk of hospital admission or death from 6.3 to 2.9%, $p=0.004$ (26). Therefore, the limited impact on hospital admission or death in our study should be interpreted in the context of this trial."*

9. Speaking of sample size, the sample size on pdf page 53 OF THE CONV-ERT in the supplement expected a 50% reduction in hospitalization. This was a large difference, potentially greater than could be expected.

Reply

One of the most important reasons to merge data from trials was indeed to increase the power to detect smaller effects than this 50% reduction. We agree that CONV-ert on its own was perhaps underpowered. The sample size of CONV-ert was calculated considering an expected 50% reduction in hospitalization following the results of the REGN-CoV-2 trial that observed a reduction of medically attended visits from 6% in the intervention arm to 3% in the placebo arm. Note that several monoclonal antibodies have shown a 70-80% reduction in hospital admission (sotrovimab, casirivimab/imdevimab). So compared with these results, the expected 50% reduction makes sense.

10. However, what is more interesting on this page is the discussion of the viral load. This data would be the mechanistic data for treatment. Why was this data not presented?

Reply

COMPILE home focused on clinically relevant outcomes (hospital admission, symptom duration, ICU admission) that were available in all patients enrolled in the participating trials. Therefore, viral loads were not part of the COMPILEhome study protocol (see supplementary protocol file) and were not collected in the CoV-Early study.

Furthermore, while an effect on viral load excretion can be expected to correlate with a better outcome, the opposite is not true. Indeed, even without any effect of an antiviral treatment on the nasopharyngeal viral load level, it can still have an effect on the viral load in the lungs as well as an important clinically relevant effect as well. Indeed, while remdesivir decreased hospital stay and in a more recent study also decreased the risk for hospital admission by 87% ($p<001$), no significant difference in nasopharyngeal viral load was observed (<https://www.nejm.org/doi/full/10.1056/NEJMoa2116846>)

Reviewer #4 (Remarks to the Author): Statistical expert comments

1. It's important for studies of convalescent plasma to be reported so that in aggregate they can inform the medical community about how efficacy might differ by timing, quality of CP, patient characteristics etc. The challenges of doing randomized double blind CP trials were formidable and it's a credit that these trials were launched and completed. Pooling ongoing trials as a practical remedy to counteract slow recruitment in individual is a great idea and important to circulate.

Evaluating the treatment every 6 weeks with no adjustment for multiple looks and no planned end is natural from the Bayesian viewpoint but makes it difficult to understand what the strength of evidence really is, especially so for a medical audience. Since this paper reports a null result, this concern is not critical, but had the study showed a benefit, it would be important to understand the frequentist operating characteristics of this approach. Some complexities include the 2 related guidelines for stopping (one involving the PO and logistic model, the other the PO model) and the monitoring every 6 weeks which could end up being many looks between few patients for a long trial with slow accrual. Such a cadence could meaningfully increase the chance of a false positive.

The study stopped because of dwindling accrual and a changing COVID landscape. It would be good to report the planned sample size for each study to get a feel for the fraction of planned information that was achieved. It is a little complicated though because you did pool the two trials so if each trial achieved 50% of the planned sample size, maybe that would be close to the planned information.

Reply:

We would like to thank the reviewer for the time taken to review our manuscript and for his/her feedback. The planned sample size for each of the individual trials are mentioned in the results section under trial profile. It was 690 and 474 respectively meaning that both reached well over 50% of planned sample size (59% and 79% respectively) and that the combined data delivers the planned amount of information.

2. Another approach would be to do a post-hoc power calculation that used the lumped data to estimate nuisance parameters and then report what treatment effect could be detected with posterior probability of 80% or 90% based on a single analysis of the entire dataset.

Reply:

We agree and we are grateful for this comment. As the assumptions about the distribution across the 5 ordinal scale categories turned out to be somewhat different (fewer deaths and hospital admissions in particular), we therefore have performed the suggested post-hoc power calculation with the actual distribution that we observed for the control patients in the combined data.

Because the COMPILHome methodology and analysis plan of the primary endpoint is nearly identical to the CoV-Early protocol we can refer to page 35 of the CoV-Early protocol where this is provided (for more details see the CoV-Early protocol available as supplementary data) "*Assuming a common odds ratio of 0.63, using a two-sided test in a proportional odds model, assuming 15% attrition (loss to follow up and admission events occurring before inclusion) a sample size of 690 is needed for a study with a power of 80% (923 for 90% power)*". Note that this calculation was performed in a frequentist way but was verified with extensive simulation to also apply to our Bayesian approach with continuous monitoring.

With the observed distribution in the control patients over the 5 categories, the effect size that was detectable with 80% power given the 782 evaluable patients turned out very similar and slightly smaller than the planned effect size in the CovEarly trial (OR 0.65 was already detectable instead of OR of 0.63 as targeted by CovEarly).

Given the raw indication of both trials delivering more than 50% of planned information and this more formal post-hoc power calculation we believe the combined COMPILHome study was not under-powered.

Hence, we have added this paragraph in the discussion: *“As our assumptions about the outcome across the ordinal scale were somewhat different than anticipated in the original sample size calculation (fewer hospitalizations and deaths in particular), we repeated the calculation of the effect size that our study was powered for post-hoc. This showed that our study still had 80% power to detect an odds ratio of 0.65 for the primary endpoint, very close to the original power calculation.”*

3. You report skeptical standard deviations of 0.4, 0.5 or 0.14. Is there a way to better convey what that means to a medical audience?

Reply:

We have added the more familiar term (conservative) next to the word skeptical to the text: *“The model included a main treatment effect shared among the trials (using a skeptical (i.e. conservative) standard deviation of 0.4)”*

4. In the abstract could say the endpoint is death or hospitalization by 28 days.

Reply:

Thank you for making us aware of this. We have added this to the abstract and now it reads as follows: *“The two primary endpoints were (a) a 5-point disease severity scale (fully recovered by day 7 or not, hospital or ICU admission and death by 28 days) and (b) a composite of hospitalization or death by 28 days”.*

5. You write the standard deviation for the covariates was 0.5, really that’s the standard deviation for the regression coefficient.

Reply:

We corrected this in the text which now reads: *“The following covariates were included with a standard deviation of 0.5 for the prior distribution of their effects: ...”*

6. For the log-rank test, how were the deaths treated? Presumably censored at day 29. However, if a patient enrolled on June 5 died on June 10 and the data was frozen on June 15, then that patient should be censored at day 10 post-randomization, i.e. the Fine-Grey approach. I suspect you have no such patients so censoring all deaths at day 29 would be fine.

Reply:

We thank the reviewer for his/her comment. As there were only 3 deaths in total, we performed this analysis excluding these patients. However, we agree with the point of the reviewer that treating

death as a competing event is a more elegant solution, and we therefore have adjusted this analysis by including the 3 subjects that died during follow-up and then censoring them at day 29.

7. I'm not sure that your pseudo-code corresponds to a Dirichlet distribution with a one vector for the parameters.

This was indeed an oversight. Thanks for noticing this!

We expected the code used Dirichlet, but it turned out to use a very weakly informative prior based on wide t-distributions. We decided not to make any changes in the final analysis anymore at this point and have now explained this discrepancy in an additional paragraph 5 in the statistical analysis plan (re-submitted together with the answer to the reviewers). For reassurance we did recalculate the results with the Dirichlet (using `blrm()` function). We found the results to be highly similar.

EDITORIAL REQUESTS:

Regarding the points raised by the Editorial team, we would like to note the following:

- We now provide the requested policy and forms for resubmission.
- We have moved the Methods section to the end as requested.
- We have added a "Data Availability" and a "Code Availability" sections.
- We have added the authors' names to each of the Consortia (CoV-Early study team and CONV-ert study team).
- In the attached revised manuscript, we have also solved some typo errors (highlighted).

Reviewers' Comments:

Reviewer #3:

None

Reviewer #4:

Remarks to the Author:

Thanks for the responses.